# A Load–Velocity Relationship in Sprint?

**DOI:** 10.3390/jfmk8030135

**Published:** 2023-09-15

**Authors:** Roland van den Tillaar, Sam Gleadhill, Pedro Jiménez-Reyes, Ryu Nagahara

**Affiliations:** 1Department of Sports Sciences, Nord University, 7600 Levanger, Norway; 2University of South Australia, Adelaide SA 5000, Australia; sam.gleadhill@unisa.edu.au; 3Centre for Sport Studies, Rey Juan Carlos University, 28943 Madrid, Spain; pedro.jimenezr@urjc.es; 4Faculty of Sports and Budo Coaching Studies, National Institute of Fitness and Sports in Kanoya, Kanoya 891-2393, Japan; nagahara@nifs-k.ac.jp

**Keywords:** force–velocity, resisted sprints, assisted sprints, laser, robotic pulley system

## Abstract

The aims were to compare predicted maximal velocity from load–velocity relationships established with different resisted and assisted loads by different regression analyses to the measured maximal velocity during sprint running, and to compare maximal velocity measured between a robotic pulley system and laser gun. Sixteen experienced male sprinters performed regular 50 m sprints, a 50 m with 5-kilogram-assisted sprint, and 10, 20, 30, and 30 m resisted sprints with, respectively, 65, 50, 25, and 10% calculated reduction in maximal velocity. Maximal velocity obtained by laser gun during the regular sprint was compared with predicted maximal velocity calculated from four trendlines (linear and polynomial based upon four resisted loads, and linear and polynomial based upon four resisted and one assisted load). Main findings demonstrate that the robotic pulley system and laser measure similar maximal velocities at all loads except at the load of 10% velocity reduction. Theoretical maximal velocity based upon calculated predictions were underestimated by 0.62–0.22 m/s (2.2–0.78 km/h; 6.7–2.3%) compared to measured maximal velocity. It was concluded that different regression analyses underestimated measured maximal velocity in regular sprinting and polynomial regression analysis (with resisted and assisted loads) estimation was closest to measured velocity (2.3%).

## 1. Introduction

An individual’s one repetition maximum (1-RM) can be measured directly and used to prescribe intensity in subsequent resistance training programs. However, there are many practical limitations to direct 1-RM measurement including injury risks, time constraints, measurement frequency, and participant preparedness [1,2,3]. Thus, often a load–velocity relationship based on velocity achieved at different submaximal loads is established to predict the 1-RM for an exercise [4]. Load–velocity relationships have been used to accurately predict 1-RM for popular free weight and Smith machine resistance exercises such as the bench press [4,5,6], with many interventions now based on these predictions to prescribe load.

Recently load–velocity relationships have been further applied to resisted sprint training using loaded sled towing, demonstrating a more individualised approach to prescribing load compared to using a percentage of body weight [7]. The relationship of load–velocity and force–velocity demonstrates the magnitude to which incrementally increasing load results in a decrease in velocity during resisted sprinting [7,8]. Once established, an individual’s load–velocity profile may be used to prescribe load based on the predicted amount of velocity decrement (from a measured maximal velocity). Using this velocity decrement method, coaches may manipulate loads to target different training zones in the force–velocity curve or different phases of a sprint. Previous research suggests that heavier loads may be most beneficial to improve initial acceleration capability, while lighter loads are more specific for the transition (later acceleration) to maximal velocity phase [9,10]. In theory, the load–velocity relationship may extend to both ends of the spectrum, thus, it may be possible to predict the maximal sprint velocity (when velocity is maximal and load is zero) or maximal load achievable (when velocity is lowest and load is greatest) with multiple loaded trials alone. However, no known research elucidates whether regression established with multiple loads alone extends to the outer ranges of a load–velocity profile, which may better the understanding of this relationship and have practical training applications.

In addition to resisted sled towing, there are now several robotic pulley machines, such as the 1080 Sprint^TM^ (1080 Motion, Lidingö, Sweden) and dynaSpeed (Ergotest Technology AS, Stathelle, Norway), capable of producing accurate resistance or assistance when sprinting, with the added capability of applying variable resistance within a single sprint repetition. These devices are further capable of measuring force, velocity, and power during sprint trials, thus, can be used to develop load–velocity profiles [11]. However, it is not known if these machines are as accurate as previously validated equipment such as the laser gun [12]. Fornasier-Santos et al. [13] and Rakovic, Paulsen, Helland, Haugen, and Eriksrud [11] found that the 1080 Sprint^TM^ was accurate in relation to the laser gun in measuring acceleration and sprint velocity. However, it is not known how accurate the dynaSpeed is. Furthermore, it is not known if sprinting with resistance could predict the maximal velocity without resistance.

Therefore, the aim of this study was twofold, firstly to compare the predicted maximal velocity from load–velocity relationships established with different resisted and assisted loads by different regression analyses (linear and second order polynomial) to the absolute measured maximal velocity during 50 m sprints for experienced male sprinters. Secondly, a comparison between the maximal velocity measured with the robotic pulley system (dynaSpeed) and laser gun was investigated to make it possible to use both systems independently and to investigate if it is possible to use both systems interchangeably.

## 2. Materials and Methods

To investigate the load–velocity relationship and compare it with the maximal velocity during 50 m sprints, both linear and 2nd order polynomial trendlines were calculated based upon the maximal velocity from the different resisted and assisted sprints, as previous studies [4,5,6] have shown linear relationships between load and velocity in strength exercises, while the force–velocity relationship on muscle level by Hill [14] and the load–velocity relationship in overhead throwing [15] were a polynomial one. Furthermore, the relationship with and without assisted sprints was investigated, as not everybody has the opportunity to include assisted sprints, when, for example, using sleds that only can give resistance to establish a relationship and to predict maximal sprint velocity for maximal sprints. Furthermore, it would be important to assess the maximal velocity estimation from the four points derived from the loaded conditions as well as to elucidate whether the addition of the assisted load (i.e., five points) could enhance the reliability of the load–velocity relationship in order to estimate the maximal velocity.

### 2.1. Participants

Sixteen experienced male sprinters (age: 20.5 ± 1.4 years; stature: 1.75 ± 0.06 m; body mass: 68.3 ± 6.4 kg; 100 m personal best, 11.31 ± 0.38 s) belonging to a university athletic club volunteered to participate in the current study. Their training experiences as sprinters were 7.6 ± 1.7 years. Moreover, they performed 2.5 to 3 h of training 5 days per week at the time of the experiment. They were instructed to avoid undertaking any resistance training targeting their lower body in the 48 h prior to testing. Each participant was informed of the testing procedures and possible risks, and written consent was obtained prior to the study. The study complied with current ethical regulations for research, was approved by the local ethics committee, and conformed to the latest revision of the Declaration of Helsinki.

### 2.2. Procedure

Each participant performed three sessions: two familiarisation sessions and one test session with at least two days between each. In each session, each participant conducted an individualised warm-up. In the first familiarisation session, each participant performed two 50 m sprints followed by four 30 m sprints with, respectively, 10% (6–8 kg) and 25% (15–17.5 kg) body mass resistance. After that, two 20 m sprints with 40% (25–30 kg) body mass resistance were performed, followed by two assisted 50 m sprints with, respectively, 4 and 5 kg pulling assistance. Between the 50 m sprints a 5 min rest was prescribed, while for the shorter distances the participants had 3 min rest between each run. The best of each of the resisted sprints with the different % of body mass were used to calculate with a linear regression relationship a load–velocity of 10, 25, 50, and 65% reduction in maximal velocity for the next familiarisation and test sessions.

In the second familiarisation session, the athletes started again with a regular 50 m sprint, followed in a stratified random order for each athlete with a 50 m with 5-kg-assisted sprint, and a 10, 20, 30, and 30 m sprint with, respectively, 65, 50, 25, and 10% calculated reduction in maximal velocity. These different loads were chosen to obtain a view over the whole spectrum of the load–velocity relationship. Rest between each sprint was at least 5–6 min.

During the test session, the same protocol was used as in the second familiarisation session, in which after the normal 50 m sprint, half of the athletes started with the 50 m assisted sprint, while the other half started with the resisted sprints in random order. This was done to avoid the effect of fatigue and due to swapping the equipment from assisted to resisted and sprinting direction. Participants wore their own spiked sprint race shoes, and every sprint was performed on an indoor athletic track.

The active resistance and assistance were provided by dynaSpeed (Ergotest Technology AS, Stathelle, Norway) with the possibility of 0.1 kg increments in loads. Participants initiated each sprint in the resisted sprints from a three-point start and a standing start when performing assisted sprints behind a line taped on the floor. Distance over time measurements were recorded continuously during each attempt using a CMP3 Distance Sensor laser gun (Noptel Oy, Oulu, Finland), sampling at 2.56 KHz. Velocity was automatically calculated as part of the Musclelab system (Ergotest Technology AS, Statthelle, Norway). The dynaSpeed and laser gun were placed, respectively, 4 and 5 m behind the start of the participant during the resisted sprints, while during the assisted sprints dynaSpeed was placed at 82 m from the start to give the participant enough distance to stop after the assisted sprint.

During a pilot study, three athletes ran for 10 m with different loads varying from 5–40 kg with 5 kg increments with a wireless force sensor of 200 kg capacity and 200 Hz sampling rate (Ergotest Technology AS, Statthelle, Norway) to calculate the exact pulling loads (N). It was found that the pulling loads were on average 4.4% higher than the actually loads. The Pearson correlation between actual load and measured load was 0.999 and the formula was:Measured load (N) = 9.99 × pulling load set (kg) + 3.5132

This formula was then used to calculate exact pulling loads (N) for the different loads. Linear and second order polynomial trendlines were calculated from the maximal velocities measured with the laser and dynaSpeed based upon the four different resisted loads and the four resisted and one assisted load together with the coefficient of determination (*r*^2^).

### 2.3. Statistical Analysis

The maximal velocities obtained by the laser gun and dynaSpeed were compared with a 2 (equipment) × 5 (load: assisted and resisted loads) model for analysis of variance (ANOVA) with repeated measures. To compare the maximal velocity during the 50 m obtained from the laser gun with the predicted maximal velocity calculated from the four trendlines (linear and polynomial trendline based upon the four resisted sprint loads and linear and polynomial trendlines based upon the resisted and assisted sprint loads), a 2 (equipment) × 4 (calculation method) model for ANOVA with repeated measures was performed. Furthermore, a 2 (equipment) × 4 (calculation method) ANOVA on the *r*^2^ was used to investigate if coefficient of determination changes with the different calculation methods. Holm–Bonferroni post hoc comparisons were performed for pairwise comparisons between the different loads and calculation methods. The level of significance was set at *p* < 0.05, and all data are expressed as mean ± standard deviation (SD). Analysis was performed with SPSS Statistics for Windows, version 25.0 (IBM Corp., Armonk, NY, USA). Effect size was evaluated with Eta partial squared (η_p_^2^) where 0.01 < η_p_^2^ < 0.06 constituted a small effect, 0.06 < η_p_^2^ < 0.14 a medium effect, and η_p_^2^ > 0.14 a large effect [16].

## 3. Results

The used loads gave a reduction in maximal velocity of approximately 89, 74, 51, and 40% of the maximal velocity obtained during the unloaded 50 m sprint (Table 1). No significant effect of equipment (F = 0.31, *p* = 0.59, η_p_^2^ = 0.02) was found. However, a significant interaction of equipment*load effect (F = 3.31, *p* = 0.016, η_p_^2^ = 0.18) was found. Post hoc comparison shows a significant (*p* = 0.024) lower maximal velocity measured with the dynaSpeed at 10% reduction load compared to the laser gun (Table 1).

All the different load–velocity trendline calculations underestimated the maximal sprint velocity at the free 50 m sprint (F = 88.8, *p* < 0.001, η_p_^2^ = 0.86). Only a significantly lower maximal sprint velocity with the dynaSpeed was calculated compared to the laser gun when using the linear four-point trendline (Figure 1). In addition, with dynaSpeed, the calculated velocity between each of the calculation methods was significantly different, while calculated velocity measured with the laser was only significantly lower when calculating it with the linear four-point trendline compared to the other three calculation methods (Figure 1).

The coefficient of determination (*r*^2^) was significantly different between the calculation methods (F = 39.6, *p* < 0.001, η_p_^2^ = 0.73), but not significantly different between the measuring equipment (F = 3.77, *p* = 0.071, η_p_^2^ = 0.20). Also, a significant interaction effect was found (F = 6.7, *p* = 0.007, η_p_^2^ = 0.31). Post hoc comparison shows that the significantly lowest *r*^2^ is found with the linear trendline based upon five points, followed by the linear four-point method with both laser and dynaSpeed. The highest *r*^2^ was found in both polynomial trendline calculations with no significant difference between these two. In addition, a significantly lower r^2^ was found with the linear five-point trendline calculation method when measuring with dynaSpeed compared with the laser gun (Figure 2).

## 4. Discussion

The aims of this study were to compare the predicted maximal velocity from load–velocity relationships established with different resisted and assisted loads by different regression analyses to the absolute maximal velocity during 50 m sprints for experienced male sprinters and to compare maximal velocity measured with the robotic pulley system and laser gun. The main findings of this study were that the dynaSpeed and laser measured similar maximal velocities at all loads except at 10% reduction in maximal velocity. Furthermore, the theoretical maximal velocity based upon the different calculation models were underestimated (0.62–0.2 m/s; 6.7–2.3%) compared to the measured maximal velocity during the normal 50 m sprint. By using a polynomial model with resisted and assisted sprints, the estimation became the closest to the measured velocity.

The dynaSpeed system measured lower maximal velocity at the 10% reduction load compared with the laser system, which could be mainly caused by the measurement and calculation method. With the dynaSpeed, the velocity is measured by the rotations of the wheel on which the cable rotates, while with the laser, the distance is instantly measured over time with a much higher sampling frequency. It is probably due to the different filtering of the distance data and fitting of the velocity curve over the data that different maximal velocities were found. However, the difference was only 0.4%, which, in a practical sense, is almost nothing.

However, due to the lower velocities measured with the dynaSpeed at 10% reduction velocity load, this resulted in lower predicted maximal velocity for the unloaded 50 m sprint compared with the laser gun, especially for the linear four points calculation method.

However, the calculated predicted maximal velocity between the two systems was very low: 0.08 m/s when using four points (linear) and only 0.04 m/s when using five points (polynomial). Thereby, we could confirm that both systems provide practical information to monitor speed capabilities from several loaded conditions when sprinting. Thus, it could be safely argued that the observed experimental findings do not provide reasons for expecting that load–velocity relationships calculated from one system or another would prominently be different if a range of loads is considered for calculations.

All predicted maximal velocities show very strong correlations with the measured maximal sprinting velocity, highlighting that the sprint exercise with different loads (commonly known as sled training) is in line with findings observed by Cahill et al. [17] with young sprinters using sled sprint. Thus, it could be added to the wide list of basic resistance training exercises (e.g., back squat, bench press, bench pull, leg press, and pull-up) that have shown this linearity and the importance of considering movement velocity as a good indicator of relative load in order to prescribe training at different zones of training, as proposed by Cahill, Oliver, Cronin, Clark, Cross, and Lloyd [17] in the case of sprinting.

However, our study also reveals that the individual load–velocity relationship is better (higher coefficient of determinations) when a second order polynomial is used compared with the linear regression (Figure 2). When using only the resisted sprint results, the maximal velocity prediction is lower with a lower correlation than when using polynomial regression or/and including the assisted sprint velocity for the calculation. By using a polynomial regression, a curve is introduced, which results in a higher maximal velocity at the unloaded condition. When using the assisted sprint in the linear regression analysis, the predicted velocity increases due to the higher assisted sprint velocity, which does not follow the linear relationship as well as with the four points linear regression. As the line is forced to be linear, while the five points lay in a curvilinear matter, the correlation coefficient decreases (Figure 2) as shown by a typical example in Figure 3. However, the *r*^2^ calculated with the linear and second order polynomial regressions are all nearly perfect (*r*^2^ >0.99), but this is mainly due to the large range of loads used and few points (four and five) to establish the trendlines and *r*^2^.

The difference with the actual measured maximal velocity with the laser was in the linear regression over the four resisted sprints, at 0.54 m/s, and decreased to 0.22 m/s when using polynomial regression and/or include the assisted sprints. This is still an underestimation of at least 0.22 m/s (0.78 km/h; 2.3%) on maximal velocity, which is a large difference when taking into account that maximal velocity is one of the main factors of winning or losing in a sprint. This discrepancy between the predicted maximal velocity and actual measured velocity is probably caused by the fact that the athletes wore a belt around their waist. This belt could inhibit some movements around the hip, which are very important when sprinting at maximal velocity [18]. During maximal sprint, tilting of the pelvis can cause changes in muscle use of hamstring and quadriceps, which gives the possibility for more efficient propulsion [19]. With the belt around the waist, this tilting could be inhibited. However, a kinematic analysis should be conducted to investigate if this is happening when wearing a belt and pulling with it. A possibility to avoid this possible inhibition is by attaching the resistance by a line higher up to the body by a harness. However, this also results in a different movement pattern during resisted sprints as found by Alcaraz et al. [20], which is probably due to the fact that the attachment is too high above the centre of mass and thereby causes these changes. Future studies should investigate how the different resisted sprint loads and attachment points affect the joint kinematics during sprints to study if there is a change during maximal unloaded sprints in joint kinematics.

Another possibility for the underestimation of the measured maximal velocity with the different methods is due to the type of relationship between load–velocity. As Hill [14] shows, there is a curvilinear relationship between force–velocity for muscle properties. In resistance training, most of the time, a linear relationship for load–velocity was found when using loads varying between 5–90% of 1-RM. In, for example, overarm throwing, a linear relationship was found [21] when using balls varying from 0.2 to 0.8 kg (typical throwing ball weights). However, when using balls across a larger range (0.5–5 kg), a curvilinear relationship between load–velocity was found [15], which indicates that perhaps a larger range of loads should be included in future studies.

This study also has some limitations. Firstly, no loads were used that resulted in higher velocity decrements, due to the limitation of the motor of dynaSpeed and that the pulling force could pull the athlete backwards at the start when using higher loads. Secondly, no kinematic joint analysis was conducted that could confirm the statement about the limitation of the of hip movements due to the belt around the waist. Thirdly, only male participants were included in this study and may constrain the generalizability of the findings to female sprinters. Lastly, in the present study, only the acute effect and relationship between load–velocity during sprinting was established. Thereby, it is not possible to state if this relationship can be changed and what the effect of this change would be upon the overall sprint performance. Therefore, we suggest that in future studies, sprints with female sprinters and a larger range of loads should be performed in which also joint kinematics are measured, with perhaps the line attached at different positions of the body. Furthermore, training interventions should be conducted using these different loads to investigate what the effect of training would have upon the load–velocity relationship and upon the overall sprinting performance.

## 5. Conclusions

Based upon the findings of the study, it was concluded that when measuring with dynaSpeed and laser gun, measurements of similar maximal velocity over different loads were obtained. Furthermore, estimation of maximal velocity based upon different load–velocity calculations could be underestimating the maximal velocity during unloaded sprints varying from 0.62–0.22 m/s (2.22–0.78 km/h), while using polynomial regression analysis with results from both resisted and assisted sprints gave the lowest underestimation of maximal velocity (2.3%).

## Figures and Tables

**Figure 1 jfmk-08-00135-f001:**
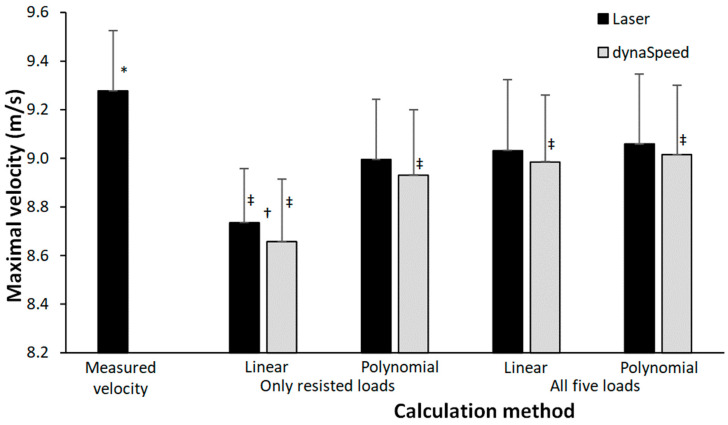
Maximal measured velocity and calculated velocity calculated by a polynomial and linear trendline based upon four points (only resisted sprints) and five points (resisted and assisted sprints). * Indicates a significant difference with all calculation methods on *p* < 0.05 level. † Indicates a significant difference between the maximal velocity obtained by laser and dynaSpeed for this calculation method on a *p* < 0.05 level. ‡ Indicates a significant difference with all other calculation methods for this equipment on a *p* < 0.05 level.

**Figure 2 jfmk-08-00135-f002:**
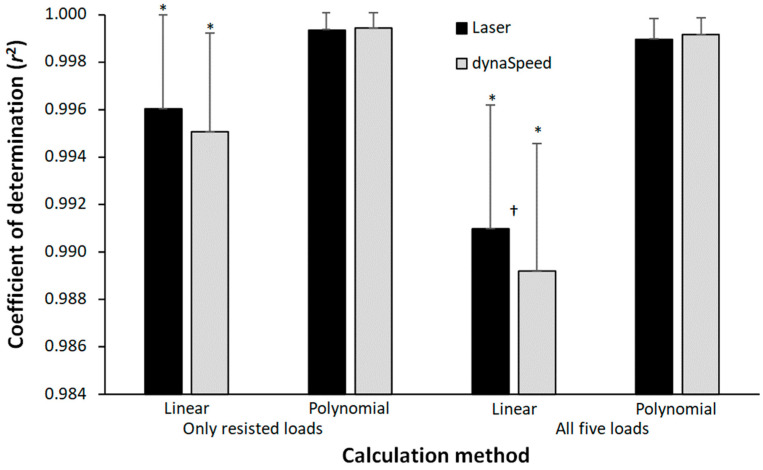
Coefficient of determination based upon four points (only resisted sprints) and five points (resisted and assisted sprints). * Indicates a significant difference with all other calculation methods on *p* < 0.05 level for this equipment. † Indicates a significant difference between the coefficient of determination obtained by laser and dynaSpeed for this calculation method on a *p* < 0.05 level.

**Figure 3 jfmk-08-00135-f003:**
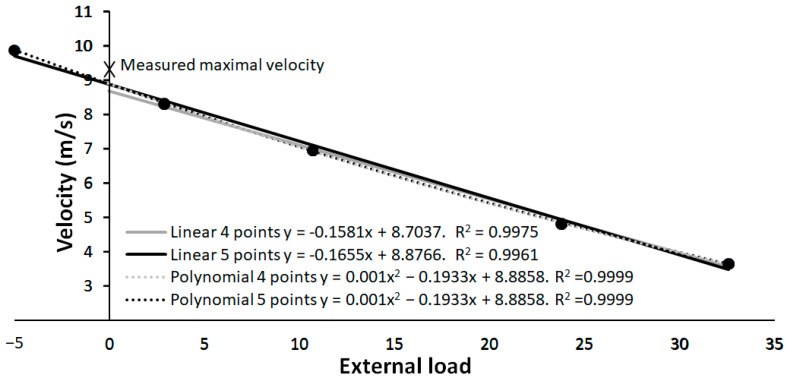
Typical example of different maximal velocities measured with laser gun at the different loads with the different linear and polynomial regression analyses.

**Table 1 jfmk-08-00135-t001:** Load and maximal measured velocity with dynaSpeed and laser gun with percentage of maximal velocity during normal 50 m sprint.

			Expected Reduction in Maximal Velocity			
	Assisted	Normal	10%	25%	50%	65%
Load (kg)	5 ± 0	0	3.2 ± 0.6	10.9 ± 1.1	24.1 ± 2.3	32.0 ± 2.7
Maximal velocity						
Laser gun (m/s)	10.10 ± 0.30	9.28 ± 0.25	8.27 ± 0.22	6.86 ± 0.25	4.72 ± 0.16	3.72 ± 0.18
dynaSpeed (m/s)	10.11 ± 0.30	-	8.23 ± 0.21 *	6.84 ± 0.23	4.70 ± 0.15	3.75 ± 0.21
% of maximal velocity						
Laser gun	108.8 ± 1.2	100	89.1 ± 1.6	74.0 ± 1.9	50.9 ± 1.9	40.1 ± 2.3
dynaSpeed	109.0 ± 1.5	100	88.7 ± 1.4 *	73.7 ± 1.7	50.7 ± 1.8	40.4 ± 2.2

* indicates a significant difference between the maximal velocity obtained by laser and dynaSpeed on a *p* < 0.05 level.

## Data Availability

The data presented in this study are available on request from the corresponding author. The data are not publicly available due to national laws of the Norwegian government on privacy.

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
