# Peer review of "A Load–Velocity Relationship in Sprint?"

_jfmk, 2023, doi:10.3390/jfmk8030135_

Round 1

Reviewer 1 Report

This manuscript presents a comparative analysis of two methodologies employed for the determination of maximal velocity through load-velocity profiling in male sprinters. Overall, the study conducted by a proficient research team demonstrates a commendable level of rigor and precision. Given the burgeoning interest in sprint systems, the methodological insights provided by this study hold significant relevance and are poised to captivate a wide readership, particularly among practitioners utilizing such systems. It is noteworthy that the study participants are experienced sprinters, further enhancing the credibility of the research.

While the current manuscript is generally well-structured, a few minor clarifications and inquiries for the authors' consideration are as follows:

1. Could the authors provide more comprehensive information regarding the training history and experience of the participants? This additional detail would bolster the confidence in the participants' training backgrounds.

2. Consider utilizing mathematical notation in lieu of script notation for the presentation of formulas. This adjustment can contribute to enhanced readability and understanding.

3. The inclusion of a limitations section is appreciated. It would be valuable to acknowledge that the exclusive inclusion of male participants may constrain the generalizability of the findings to female sprinters. This serves to underscore the need for caution when extrapolating the results to a broader population.

4. The use of an ANOVA model for analyzing the coefficient of determination statistics warrants clarification. It prompts the question of what unique insights this analysis offers compared to a straightforward summary table of averages. While the application of ANOVA on the Y-intercept appears justified, its necessity and potential redundancy in the context of coefficient of determination statistics require further explanation. Can the authors please provide a rationale for employing this analytical technique?

Author Response

We want to thank the reviewer for the comments. We have changed the manuscript according to the comments of the reviewer and think that it is now suitable for publication.

This manuscript presents a comparative analysis of two methodologies employed for the determination of maximal velocity through load-velocity profiling in male sprinters. Overall, the study conducted by a proficient research team demonstrates a commendable level of rigor and precision. Given the burgeoning interest in sprint systems, the methodological insights provided by this study hold significant relevance and are poised to captivate a wide readership, particularly among practitioners utilizing such systems. It is noteworthy that the study participants are experienced sprinters, further enhancing the credibility of the research.

While the current manuscript is generally well-structured, a few minor clarifications and inquiries for the authors' consideration are as follows:

  1. Could the authors provide more comprehensive information regarding the training history and experience of the participants? This additional detail would bolster the confidence in the participants' training backgrounds.

We have included more information about the training stratus of the participants in the text now.

  1. Consider utilizing mathematical notation in lieu of script notation for the presentation of formulas. This adjustment can contribute to enhanced readability and understanding.

The script notation is ok according to us for the readability.

  1. The inclusion of a limitations section is appreciated. It would be valuable to acknowledge that the exclusive inclusion of male participants may constrain the generalizability of the findings to female sprinters. This serves to underscore the need for caution when extrapolating the results to a broader population.

We have already included a paragraph of limitations (line 290-301). We have added the part of using also female sprinters to enhance generalization.

  1. The use of an ANOVA model for analyzing the coefficient of determination statistics warrants clarification. It prompts the question of what unique insights this analysis offers compared to a straightforward summary table of averages. While the application of ANOVA on the Y-intercept appears justified, its necessity and potential redundancy in the context of coefficient of determination statistics require further explanation. Can the authors please provide a rationale for employing this analytical technique?

We used an ANOVA model for analyzing the coefficient of determination to determine if correlation significantly would change when using the four different methods. Even when the correlation already was very high a significant increase or decrease would indicate a better correlation and better prediction over the different points. Since only 4 and 5 points were used for the different correlation models the correlations were already high. However, by including this extra analysis you could see that the use of polynomial correlations increased the relation between the loads and velocity, which indicates that the relationship perhaps is not linear, but more polynomial (2nd order), which you have to account for.

Reviewer 2 Report

Manuscript:

A load-velocity relationship in sprint?

Comments to authors:

The present study aims to compare predicated maximal velocity relationship using different laser and pulley system. Overall the paper is well written and concise. There are a few minor concerns discussed in each section of the manuscript.

Introduction:

Line 28-31: This sentence seems too long to follow easily. I would suggest splitting the sentence in into two. Possibly starting 2nd sentence with ‘However, there are many …’

Materials and Methods:

Line 100-101: The authors state that 10% is 6-8 kg or 25% is 15-17.5 kg. Does this imply that the system can be incremented by values of 0.5 kg - what is the resolution of the system? Please specify.

Line 110. ‘… a 10, 20, 30 m with respectively 65. 50, 25, 10% …’ These are three distances with four resistance values? Should this not be equal? * Also given in Abstract.

Line 126: ‘resp.’ - Respectively? Just write it out.

Discussion:

Line 215-216. ‘Probably due to different filtering …’ Do the authors know that the filtering is different? Please elaborate on this?

Line 247: ‘so good as with’. Change to ‘as good as with’

Figure 3: Please confirm whether polynomial 4 point and 5 point equations are exactly the same?

Line 279: ‘f.e’? Please clarify what this means.

Author Response

We want to thank the reviewer for the comments. We have changed the manuscript according to the comments of the reviewer and think that it is now suitable for publication.

Comments to authors:

The present study aims to compare predicated maximal velocity relationship using different laser and pulley system. Overall the paper is well written and concise. There are a few minor concerns discussed in each section of the manuscript.

Introduction:

Line 28-31: This sentence seems too long to follow easily. I would suggest splitting the sentence in into two. Possibly starting 2nd sentence with ‘However, there are many …’

 Changed now according to the comment of the reviewer.

Materials and Methods:

Line 100-101: The authors state that 10% is 6-8 kg or 25% is 15-17.5 kg. Does this imply that the system can be incremented by values of 0.5 kg - what is the resolution of the system? Please specify.

In the familiarization session we used these load ranges. However, during the following familiarization and final tests we used increments with 0.1 kg. This is now added to the text.

Line 110. ‘… a 10, 20, 30 m with respectively 65. 50, 25, 10% …’ These are three distances with four resistance values? Should this not be equal? * Also given in Abstract.

 We have rewritten it in a 10, 20, 30, 30 m with respectively 65, 50, 25, 10% to avoid confusion.

Line 126: ‘resp.’ - Respectively? Just write it out.

 Changed now in respectively

Discussion:

Line 215-216. ‘Probably due to different filtering …’ Do the authors know that the filtering is different? Please elaborate on this?

The filtering of the distance over time between the two systems is probably done differently, since the laser gun is sampling at 2.56Khz and dynaSpeed with another sampling rate. The filtering is probably different. However, we don’t know the exact filtering as it is automatically done in the software, which we don’t have insight of.

Line 247: ‘so good as with’. Change to ‘as good as with’

 Changed now.

Figure 3: Please confirm whether polynomial 4 point and 5 point equations are exactly the same?

In this typical example they are exactly the same, correct. However, this was not the case in all subjects.

 Line 279: ‘f.e’? Please clarify what this means.

For example. We have written it fully out now.